# Interpretability Through Invertibility: A Deep Convolutional Network With Ideal Counterfactuals And Isosurfaces

## Abstract

Current state of the art computer vision applications rely on highly complex models. Their interpretability is mostly limited to post-hoc methods which are not guaranteed to be faithful to the model. To elucidate a model's decision, we present a novel interpretable model based on an invertible deep convolutional network. Our model generates meaningful, faithful, and ideal counterfactuals. Using PCA on the classifier's input, we can also create "isofactuals"– image interpolations with the same outcome but visually meaningful different features. Counter- and isofactuals can be used to identify positive and negative evidence in an image. This can also be visualized with heatmaps. We evaluate our approach against gradient-based attribution methods, which we find to produce meaningless adversarial perturbations. Using our method, we reveal biases in three different datasets. In a human subject experiment, we test whether non-experts find our method useful to spot spurious correlations learned by a model. Our work is a step towards more trustworthy explanations for computer vision. For code: `https://anonymous.4open.science/r/ae263acc-aad1-42f8-a639-aec20ff31fc3/`

## 1 Introduction

The lack of interpretability is a significant obstacle for adopting Deep Learning in practice. As deep convolutional neural networks (CNNs) can fail in unforeseeable ways, are susceptible to adversarial perturbations, and may reinforce harmful biases, companies rightly refrain from automating high-risk applications without understanding the underlying algorithms and the patterns used by the model.

Interpretable Machine Learning aims to discover insights into how the model makes its predictions. For image classification with CNNs, a common explanation technique are saliency maps, which estimate the importance of individual image areas for a given output. The underlying assumption, that users studying local explanations can obtain a global understanding of the model (Ribeiro et al., 2016), was, however, refuted. Several user studies demonstrated that saliency explanations did not significantly improve users' task performance, trust calibration, or model understanding (Kaur et al., 2020; Adebayo et al., 2020; Alqaraawi et al., 2020; Chu et al., 2020). Alqaraawi et al. (2020) attributed these shortcomings to the inability to highlight global image features or absent ones, making it difficult to provide counterfactual evidence. Even worse, many saliency methods fail to represent the model's behavior faithfully (Sixt et al., 2020; Adebayo et al., 2018; Nie et al., 2018). While no commonly agreed definition of faithfulness exists, it is often characterized by describing what an unfaithful explanation is (Jacovi & Goldberg, 2020). For example, if the method fails to create the same explanations for identically behaving models.

To ensure faithfulness, previous works have proposed building networks with interpretable components (e.g. ProtoPNet (Chen et al., 2018) or Brendel & Bethge (2018)) or mapping network activations to human-defined concepts (e.g. TCAV (Kim et al., 2018)). However, the interpretable network components mostly rely on fixed-sized patches and concepts have to be defined *a priori*.

Here, we argue that explanations should neither be limited to patches and not rely on a priori knowledge. Instead, users should *discover* hypotheses in the input space themselves with *faithful* counterfactuals that are ideal, i.e. samples that exhibit changes that directly and exclusively correspond

to changes in the network's prediction (Wachter et al., 2018). We can guarantee this property by combining an invertible deep neural network $z = \varphi(x)$ with a linear classifier $y = w^T\varphi(x) + b$. This yields three major advantages: 1) the model is powerful (can approximate any function Zhang et al. (2019)), 2) the weight vector $w$ of the classifier directly and *faithfully* encodes the feature importance of a target class $y$ in the $z$ feature space. 3) Human-interpretable explanations can be obtained by simply inverting explanations for the linear classifier back to input space.

As a local explanation for one sample $x$, we generate ideal counterfactuals by altering its feature representation $z$ along the direction of the weight vector $\tilde{z} = z + \alpha w$. The logit score can be manipulated directly via $\alpha$. Inverting $\tilde{z}$ back to input space results in a human-understandable counterfactual $\tilde{x} = \varphi^{-1}(z + \alpha w)$. Any change orthogonal to $w$ will create an *"isofactual"*, a sample that looks different but results in the same prediction. While many vectors are orthogonal to $w$, we find the directions that explain the highest variance of the features $z$ using PCA. As the principal components explain all variance of the features, they can be used to summarize the model's behavior globally.

We demonstrate the usefulness of our method on a broad range of evaluations. We compared our approach to gradient-based saliency methods and find that gradient-based counterfactuals are not ideal as they also change irrelevant features. We evaluated our method on three datasets, which allowed us to create hypotheses about potential biases in all three. After statistical evaluation, we confirmed that these biases existed. Finally, we evaluated our method's utility against a strong baseline of example-based explanations in an online user study. We confirmed that participants could identify the patterns relevant to the model's output and reject irrelevant ones. This work demonstrates that invertible neural networks provide interpretability that conceptually stands out against the more commonly used alternatives.

## 2 METHOD

Throughout this work, we rely on the following definitions, which are based on Wachter et al. (2018):

**Definition 2.1** (Counterfactual Example). Given a data point $x$ and its prediction $y$, a *counterfactual example* is an alteration of $x$, defined as $\tilde{x} = x + \Delta x$, with a altered prediction $\tilde{y} = y + \Delta y$ where $\Delta y \neq 0$. Samples $\bar{x}$ with $\Delta y = 0$ are designated *"isofactuals"*.

Almost any $\Delta x$ will match the counterfactual definition, including those that *additionally* change aspects which are unrelated to the model's prediction, e.g. removing an object but also changing the background's color. It is desirable to isolate the change most informative about a prediction:

**Definition 2.2** (Ideal Counterfactual). Given a set of unrelated properties $\xi(x) = \{\xi_i(x)\}$, a sample $\tilde{x}$ is called *ideal* counterfactual of $x$ if all unrelated properties $\xi_i$ remain the same.

The following paragraphs describe how we generate explanations using an invertible neural network $\varphi \colon \mathbb{R}^n \mapsto \mathbb{R}^n$. The forward function $\varphi$ maps a data point $x$ to a feature vector $z = \varphi(x)$. Since $\varphi$ is invertible, one can regain $x$ by applying the inverse $x = \varphi^{-1}(z)$. We used the features $z$ to train a binary classifier $f(x) = w^T z + b$ that predicts the label $y$. In addition to the supervised loss, we also trained $\varphi$ as a generative model (Dinh et al., 2016; 2015) to ensure that the inverted samples are human-understandable.

**Counterfactuals** To create a counterfactual example $\tilde{x}$ for a datapoint $x$, we can exploit that $w$ encodes feature importance in the $z$-space directly. To change the logit score of the classifier, we simply add the weight vector to the features $z$ and then invert the result back to the input space: $\tilde{x} = \varphi^{-1}(z + \alpha w)$. Hence, for any sample $x$, we can create counterfactuals $\tilde{x}$ with an arbitrary change in logit value $\Delta y = \alpha w^T w$ by choosing $\alpha$ accordingly. Figure 1a shows several such examples. Since the generation ($\varphi^{-1}$) and prediction ($\varphi$) are performed by the same model, we know that $\tilde{x}$ will correspond exactly to the logit offset $\alpha w^T w$. Consequently, $\tilde{x}$ is a *faithful* explanation.

To show that our counterfactuals are ideal, we have to verify that no property unrelated to the prediction is changed. For such a property $\xi(x) = v^T z$, $v$ has to be orthogonal to $w$.[1] As the unrelated property $\xi$ does not change for the counterfactual $\xi(\tilde{x}) = v^T(z + \alpha w) = v^T z = \xi(x)$, we know that $\tilde{x} = \varphi^{-1}(z + \alpha w)$ is indeed an *ideal* counterfactual.

---

[1] $\xi(x)$ could actually be non-linear in the features $z$ as long as the gradient $\frac{\partial \xi}{\partial z}$ is orthogonal to $w$.

**PCA Isosurface** Since users can only study a limited number of examples, it is desirable to choose samples that summarize the model's behavior well (Ribeiro et al., 2016; Alqaraawi et al., 2020). For counterfactual explanations, the change $\Delta x$ may vary significantly per example as $\varphi(x)$ is a non-linear function. As each $x$ has a unique representation $z$ in the feature space, we want to find examples describing the different directions of the feature distribution. To isolate the effect of $w$, such examples would have the same prediction and only vary in features unrelated to the prediction.

We implement this by first removing the variation along $w$ using a simple projection $z_\perp = z - (w^T z / w^T w)w$ and then applying PCA on $z_\perp$. The resulting principal components $e_1 \ldots e_m$ are orthogonal to $w$ except of the last principal component $e_m$ which has zero variance and can therefore be discarded. The principal components span a hyperplane $\alpha w + \sum_i^{m-1} \beta_i e_i$. Since all samples on this hyperplane have the same prediction (a logit value of $\alpha w^T w$), it is an *isosurface*.

As a principal component $e_i$ is a vector in the $z$-space, we can create counterfactuals for it $\varphi^{-1}(e_i + \alpha w)$ and understand how the changes of adding $w$ differ per location in the $z$-space. The $e_1, \ldots, e_{m-1}$ are sorted by the explained variance allowing to prioritize the most relevant changes in the data. As the principal components cover the whole feature distribution, understanding the effect of $w$ on them allows forming a global understanding of the model's behavior.

**Saliency maps** Saliency maps are supposed to draw attention to features most relevant to a prediction. In our case, it is most reasonable to highlight the difference between $x$ and the counterfactual $\tilde{x}$. We can measure the difference although in an intermediate feature map $h$. The saliency map of an intermediate layer can be resized to fit the input's resolution as information remains local in convolutional networks. Per feature map location $(i, j)$, we calculate the similarity measure $m_{(i,j)} = |\Delta h_{ij}| \cos(\angle(\Delta h_{ij}, h_{ij}))$. The sign of the saliency map $m$ depends on the alignment of the change $\Delta h$ with the feature vector $h$, i.e. $(\angle(\Delta h_{ij}, h_{ij}) > 0)$. The magnitude is dominated by the length of the change $|\Delta h_{ij}|$. Figure 1b presents saliency maps for the CELEBA *Attractive* label.

**Model** Our invertible network follows the Glow architecture (Kingma & Dhariwal, 2018). The network is trained to map the data distribution to a standard normal distribution. We reduce the input dimensionality of (3, 128, 128) down to (786) by fading half of the channels out with each downsampling step. When generating a counterfactual, we reuse the $z$ values out-faded from the lower layers as they correspond to small details and noise. We have 7 downsampling steps and 351 flow layers. The network has 158.769.600 parameters in total. An important design decision is that the final layer's output is not input to the linear classifier. The PCA would fail to discover meaningful directions as the $\mathcal{N}(0, I)$ prior induces equal variance in all directions. The classifier uses the output of layer 321. The layers 322-351 are optimized using the standard unsupervised flow objective. For the first 321 layers, we also train on the classifier's supervised loss (for details see Appendix A.1).

## 3 EVALUATION

We evaluated the ability to construct hypotheses about the model's behavior on three datasets and with a user study. We focused on these aspects as our method is faithful by construction, needing no empirical confirmation. Instead, we use the strong faithfulness guarantees of our model to evaluate gradient-based attribution methods.

### 3.1 HYPOTHESIS DISCOVERY

**CelebA** A claimed utility of our method is that it allows users to discover hyphotheses about the models features used for prediction. We choose CELEBA (Liu et al., 2015), a popular face dataset, because it is a challenging dataset for feature attribution: how can an abstract concept as attractiveness be linked to pixels? Additionally, it already contains annotations (e.g. make-up, accessories, hair), which makes it easier for us to accept or reject a given hypothesis about feature importance.

We especially focus on the *Attractive* class as it is unclearer what the relevant features are. The CELEBA Dataset in general and the class *attractive* in particular are ethically questionable. How can a subjective label, which depends on individual or even cultural preferences, be reduced to a binary label? Unfortunately, (Liu et al., 2015) did not state the annotation process (which is considered good practice - (Gebru et al., 2020; Geiger et al., 2020)). Furthermore, the dataset was criticized for lacking diversity (Kärkkäinen & Joo, 2019).

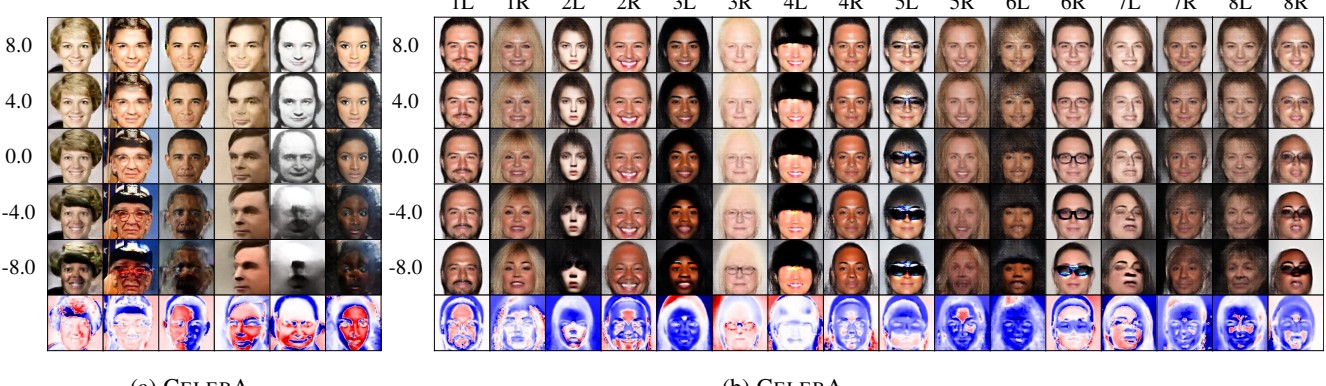

(a) CELEBA                    (b) CELEBA

Figure 1: **(a)** We generate counterfactual images by moving along the direction of the classifier weights $w$ of the *attractive* class and inverting it back to the input. Last row shows the saliency maps from the center row (logit $y=0$) to the top row ($y=8$). Blue marks features changed into a different direction and red marks features getting enhanced. **(b)** We extract principal components (L=$e_i$, R=$-e_i$) orthogonal to the classifier weight $w$. All images in a row have the exact same logit score given on the left. The saliency maps show the change between the bottom ($y=-8$) and top ($y=8$).

Figure 1b shows the first 8 principal components at different logit values. We base our investigation on them, as they cover the feature distribution well by construction. At this point, we invite the reader to study the explanations: What are your hypotheses about the model's used features?

Studying the counterfactuals in rows (3R, 5L, 6R, 8R), one might hypothesize that *glasses* influence the prediction of attractiveness negatively. To validate this, we analyzed our model's predictions on the test set. Since glasses are a labeled feature of CELEBA it is easy to test the hypothesis empirically. Only 3.5% of the portrait photos, which are showing glasses were labeled as *attractive* by the model. Furthermore, the correlation of the presence of glasses and the logit score was r=-0.35.

Another insight noticeable in 1L is that the amount and density of *facial hair* changes the prediction. The correlation of the absence of facial hair with the *attractiveness* logit score was r=0.35. At the same time, less *head hair* seemed to reduce attractiveness predictions in rows 1L, 2R, 4R. Row 6L paints the opposite picture, which illustrates the varying effect $w$ can have on different datapoints. We found a correlation ($r = 0.30$) of hair-loss (combination of baldness or receding hairline) with attractiveness.

Indicative of higher *attractiveness* appear to be a more feminine appearance (e.g. 4R in Figure 1) . This hints to a gender bias, which we confirmed as only 20.0% of men are predicted to be attractive, and the label *male* was negatively correlated with the prediction ($r = -0.59$). Further, it is noticeable that counterfactuals for higher attractiveness tend to have redder lips (1R, 2R,4R and 5L). This hypothesis could also be confirmed as the label *Wearing Lipstick* is also positively correlated ($r = 0.64$). For age, similar patterns can be found in 1L, 3R, 8L ($r = 0.44$). Table 4 in the Appendix D lists the correlation of all 40 attributes. Some attributes cannot be found in the principal components because the cropping hides them (double chin, necklace, necktie). Others describe local details such as arched eyebrows, earrings. While earrings do not show up in the counterfactuals, they are correlated with the model's logit score by r=0.20. This might be because PCA tends to capture global image features while smaller local changes are scattered over many principal components. Another explanation could be that earings are actually not that relevant: if we control for gender using partial correlation the earings are only correlated by r=-0.01.

Darker skin color seems to influence the network negatively, as in principal components (2R, 3R, 6L) a light skin color suggests high attractiveness. Since CELEBA has no labels for skin color, we annotated 3250 randomly selected images: 249 photos matched the Fitzpatrick skin type V-VI and were labeled as dark skin (Fitzpatrick, 1986). For light skin, the percentage of *Attractive* was 52.0%. The same bias is contained in the model: r=-0.187 (-0.22, -0.15)$_{95\%}$.

**Two4Two** The TWO4TWO dataset (Anonymous, 2020) is a set of computer-generated images intended to evaluate interpretable ML – to test both humans and algorithms. While the dataset is simple, we control the data generation process and can create arbitrary images to test the model. The dataset contains two abstract *animals*, Sticky and Stretchy. For Sticky, the right arms are moved inwards and for Stretchy outwards (see Figure 2b). As the arms overlap sometimes, it is beneficial also to use the color which is slightly predictive (blue for Stretchy and red for Sticky). Building

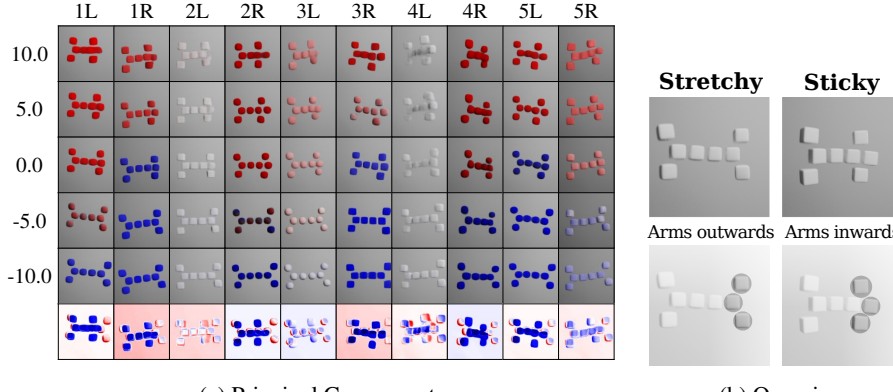

| (a) Principal Components | (b) Overview |

Figure 2: **(a)** The principal components for Two4Two. Sticky is on the top and Strechy below. The saliency maps shown below fail to highlight the object movement well. **(b)**The main feature of Stretchy are the outward moved left *arms*. For Sticky, they are moved inwards

blocks (cubes or spheres), bending, rotation, and background are sampled independently. For the Two4Two dataset, the invertible neural network $\varphi$ was only trained on an unsupervised loss, i.e. the gradients of the classifier were detachted. Probably due to the datasets simplicity, we had problems to align the unsupervised and supervised loss well.

The principal components in Figure 2a suggest that the model indeed learned to use the color bias. We an confirm this by resampling only the color and measure how the logit score is correlated: r=0.352. For the arm's position, we found a correlation with the model's probability of -0.798. Additionally, Sticky on the top seems to be more rotated, which we can also confirm as only changing the rotation results in a correlation of the logit score with the absolute value of teh rotation of with r=0.136 (0.11, $0.16)_{95\%}$. At high rotations, the model is more certain that it is a Sticky. Although not intended by the dataset, this bias can be well explained by the fact that $\varphi$ was not trained on the supervised loss.

**Black Mice** We wanted to check our method on a dataset which is not already known to have biases as the CelebA dataset and is harder for a human to understand. The Black Mice dataset Andresen et al. (2020) contains images of laboratory mice after different treatments. The label to predict is related to the amount of pain. For a detailed discussion of the dataset, see Appendix **??**. The main take-away point is that we find that the yellow bedding material, which is changed by our model's counterfactuals, is indeed predictive of the label.

## 3.2 Comparison of The Gradient of $x$ And The Directional Derivative $\mathrm{d}\varphi^{-1}/\mathrm{d}w$

In this evaluation, we propose a simple validity check for attribution methods and apply it to our method and gradient-based attribution methods. The idea is to relate saliency maps to counterfactuals. As saliency maps should highlight features most influential for the outcome of a datapoint, amplifying these features should increase the prediction and therefore create a counterfactual. We propose the following test: integrate the raw feature attribution values and then check if (1) the counterfactual increases the logit score and (2) if the changes are into the direction of $w$ or rather into the direction of unrelated properties. We measure (2) by calculating the changes in the directions of the principal components: $\boldsymbol{\xi} = E\boldsymbol{z}$ where $E$ is the matrix of all $\boldsymbol{e}_i$.

We construct an infinitesimal version of our counterfactuals by $\lim_{\alpha \to 0} \frac{\varphi^{-1}(\boldsymbol{z}+\alpha\boldsymbol{w})}{\alpha|\boldsymbol{w}|}$. This gives the directional derivative[2] of the input w.r.t. to the classifier weight: $\nabla_{\boldsymbol{w}}\boldsymbol{x} = \nabla_{\boldsymbol{w}}\varphi^{-1} = \mathrm{d}\varphi^{-1}(\boldsymbol{z})/\mathrm{d}\boldsymbol{w}$. Moving the input $\boldsymbol{x}$ into the direction $\nabla_{\boldsymbol{w}}\boldsymbol{x}$ will result in a move of $\boldsymbol{z}$ into the $\boldsymbol{w}$ direction.[3]

We evaluate the directional derivative against the raw gradient, which serves as a basis for many saliency methods (SmoothGrad, LRP$_\epsilon$, LRP$_{\alpha\beta}$, $\gamma$-rule, and integrated gradients (Smilkov et al., 2017; Bach et al., 2015; Montavon et al., 2019; Sundararajan et al., 2017)).[4] Additionally, we include SmoothGrad (*sm.g*) and build two additional methods by penalizing changes in the unrelated

---

[2] TCAV (Kim et al., 2018) uses the directional derivative of the networks output w.r.t. a concept vector $\boldsymbol{v}$: $\frac{\mathrm{d}f}{\mathrm{d}\boldsymbol{v}}$. Different to our method, TCAV computes the gradient of the forward model and not on the inverse $\varphi^{-1}$.

[3] A reader familiar with differential geometry might recognize this as the pushforward of $\boldsymbol{w}$ using $\varphi^{-1}$.

[4] The gradient and the directional derivative have a mathematical similarity which can be seen on the Jacobian: $\nabla_{\boldsymbol{x}}f = J_\varphi(\boldsymbol{x})\boldsymbol{w}$ and $\nabla_{\boldsymbol{w}}\boldsymbol{x} = J_{\varphi^{-1}}(\boldsymbol{z})\boldsymbol{w}$.

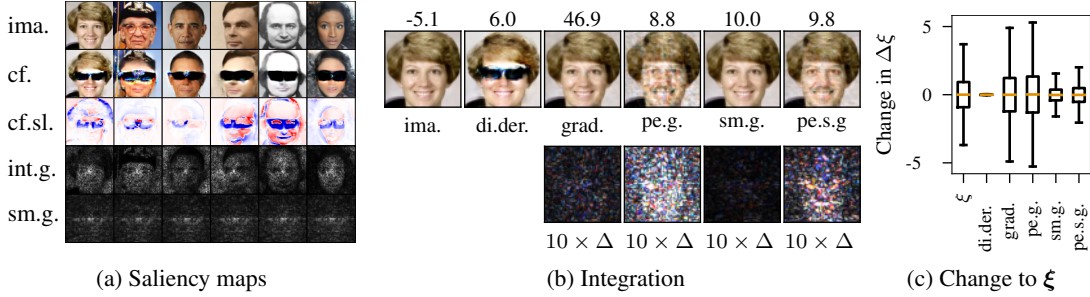

| (a) Saliency maps | (b) Integration | (c) Change to $\boldsymbol{\xi}$ |

Figure 3: **(a)** Saliency maps computed for the *Eyeglasses* class of our method (*cf.sl.*), integrated gradients (*int.g.*), and SmoothGrad (*sm.g.*). *cf.* denotes counterfactuals with logit $y = 6$. **(b)** Integration of the raw feature attribution values, e.g. gradient w.r.t. to a single neuron. The gradient (*grad*) results in a strong logit change (given on top) but fails to create visible changes. Differences with the original images (*img*) are magnified below ($\times 10$). SmoothGrad and the respective penalized version (*pe.gr* and *pe.s.g*). show similar results. The directional derivative $\mathrm{d}\varphi^{-1}/\mathrm{d}\boldsymbol{w}$ adds sunglasses. **(c)** The distribution of $\boldsymbol{\xi}$ is shown in the first row. All gradient-based methods result in strong and therefore less interpretable counterfactual. The directional derivative $\nabla_{\boldsymbol{w}}\varphi^{-1}$ changes $\boldsymbol{\xi}$ little.

properties $\boldsymbol{\xi}$ using a mean squared error with the $\boldsymbol{\xi}$ of the original image (*pe.gr.* for gradient and for SmoothGrad *pe.s.g*). The integration is done by iterative steps into the direction of the integrated quantity, e.g. for the gradient we would calculate $\boldsymbol{x}_{t+1} = \boldsymbol{x}_t + \gamma \nabla_{\boldsymbol{x}} f(\boldsymbol{x}_t)$ where $\gamma$ is a small step (see Appendix A.2 for all technical details).

Figure 3b shows exemplary results of the integration for the *Eyeglass* dimension. While the gradient-based counterfactual increases the logit score by an order of magnitude, the resulting image is hardly different from the original. Only noise patterns appear – similar to adversarial examples. SmoothGrad results in both a lower logit score and even smaller changes to the image. Penalizing changes in unrelated properties only yields amplified noise patterns. At the start of the integration, the difference in $\boldsymbol{\xi}_0$ is zero, which probably results in first moving along $\boldsymbol{\xi}$. In contrast, integrating the directional derivative adds sunglasses to the astronaut – a meaningful counterfactual.

We measure the quality of a counterfactual by measuring how strongly unrelated factors change on 100 random samples and report the results in Figure 3c. Thus, gradient-based counterfactuals do not only explain the increase of the logit score, but also all the other changes too. A user studying the gradient counterfactual could not differentiate between changes done to the prediction and the unrelated factors. The counterfactual based on the directional derivative keeps the independent factors almost unchanged up to numerical imprecision.

## 3.3 HUMAN SUBJECT STUDY

Our aim was to evaluate whether counterfactual interpolations can help lay users to form hypotheses about a models used patterns and potential biases. Evaluating explanation techniques with users is important though a challenging endeavor as it requires mimicking a realistic setting, while avoiding overburdening participants (Doshi-Velez & Kim, 2017; Wortman Vaughan & Wallach, 2020).

The choice of the dataset is important for any evaluation. Some datasets introduce participants' domain knowledge as a cofounding factor (e.g. images of dog breeds). While others like CELEBA introduce subjectivity. Datasets can have many relevant features, creating an enormous amount of possible and valid hypotheses. If participant were allowed to develop hypotheses about them without limitation this would require us to mostly evaluate them manually which would be too labor intensive. Asking participants to reason about pre-selected hypothesis prevents us from assessing their total understanding of the model as there are potentially many relevant features.

We chose the TWO4TWO data set (Section 3.1) as it addresses these issues (Anonymous, 2020). The simple scenario enables us to control the available patterns and limit the number of feasible hypotheses, allowing for comparable quantitative analysis. Concretely, we assessed a participant's judgment about the plausibility of six hypotheses. Three hypotheses were reasonable (sensitivity to spatial compositions, color, and rotation). Two others were not (sensitivity to background and shape of individuals blocks). We also asked them to reason about the model's maturity and measured their perception of the explanations using applicable statements taken from the Explanation Satisfaction Scale (Hoffman et al., 2018).

**Baseline Selection** Many studies in machine learning solely demonstrate their methods feasibility without a baseline comparison (e.g. Ribeiro et al. (2016); Singla et al. (2020)). In contrast, we carefully considered what would be *the best alternative method available* to allow users to *discover* hypotheses about a model. As discussed previously in this work, many feature attribution techniques suffer from a lack of faithfulness and fail to provide meaningful counterfactuals. If counterfactuals are meaningful and faithful to the model they can be expected to look similar. Hence, comparing our method to other counterfactual generation methods (e.g. to GANs (Singla et al., 2020)) provides limited insight about their practical usefulness if there are alternative ways of discovering similar hypotheses. As for saliency maps, in addition to concerns about their faithfulness, there are also growing concerns about their practical usefulness. While early works found they can calibrate users' trust in a model (e.g. Ribeiro et al. (2016)), more recent works cast doubts about this claimed utility (Kaur et al., 2020; Chu et al., 2020). Studies found that while they are useful to direct users' attention towards relevant features, they facilitate limited insight (Alqaraawi et al., 2020; Chu et al., 2020). Other studies found they may even harm users' understanding about errors of the model (Shen & Huang, 2020). After all, users often seem to ignore them, relying predominantly on predictions instead when reasoning about a model (Chu et al., 2020; Adebayo et al., 2020).

While we introduce a faithful saliency method, we do not claim that it would not suffer from the same usability problems, especially with lay users (see Figure 7 for examples generated for TWO4TWO). After all our maps would need to be used in conjunction with counterfactuals, potentially adding a dependent variable (presence of saliency map) to experiment. For these reasons, we decided against considering saliency maps in this evaluation.

We also did not consider methods based on infilling (e.g. Goyal et al. (2019)), as we expected them to suffer from similar usability problems. For example, as they explain features locally by removing them, paying no attention to overlapping features, they can be expected to remove the entire object from the scene when explaining the model's bias towards the object's color. This would leave the user puzzled what feature of the object (shape, position or color) is important.

A simple alternative is to study the system predictions on exemplary input. Such reasoning on natural images to understand model behavior has surfaced as a strong baseline in another study (Borowski et al., 2020). Hence, we choose example-based explanations as our baseline treatment.

**Explanation Presentation** Considering that participants' attention is limited and to allow for a fair comparison, we wanted to provide the same amount of visual information in both conditions. We choose a 30x5 image grid (3 rows shown in Figure 4). Each column represented a logit range. Ranges were chosen so that high confidence predictions for *Stretchy* were shown on the far left column and high confidence predictions *Sticky* on the far right. Less confident predictions were shown in the directly adjoining columns. The remaining middle column represented borderline-cases. This visual design had prevailed throughout numerous iterations and ten pilot studies, as it allows users to quickly scan for similar features in columns and differing features in rows.

Both conditions only varied in the images that were used to populate the grid. In the baseline, the grid was filled with images drawn from validation set that matched the corresponding logit ranges. In the *counterfactual interpolations conditions*, only the diagonal of the grid was filled randomly with such "original" images. They were marked with a golden frame. The remaining cells were filled row-wise with counterfactuals of the original images that matched the corresponding columns score range.

Our online study was preregistered [5] and followed a between-group design. Participants (N=60) were recruited from Prolific and needed to hold an academic degree with basic mathematical education. Participants were randomly but equally assigned to view either counterfactual interpolations or the baseline. Upon commencing the study on the Qualtrics platform, participants were shown handcrafted video instructions. After that, they studied the image grid while rating their agreement to six statements on a 7-point Likert scale. Participants also rated their agreement to four applicable statements taken from the Explanation Satisfaction Scale (Hoffman et al., 2018).

**Study Results and Discussion** The significance of rating difference was assessed using a Kruskal-Wallis Test. To account for multiple comparisons, we applied Bonferroni correction to all reported p-values. For a detailed assessment of all preregistered hypothesis, please refer to the Appendix (Section E.1). Figure 4a summarizes the responses.

---

[5] see supplementary material

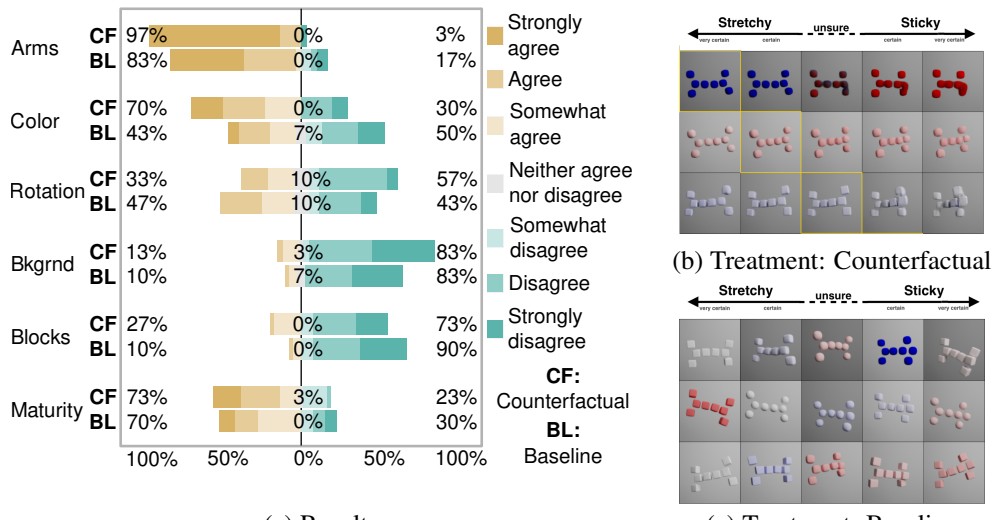

(a) Results

(b) Treatment: Counterfactual

(c) Treatment: Baseline

Figure 4: Left: Participants agreements to statements about the networks used patterns. Right: The study interface (vertically cropped) in the counterfactual interpolations (Top) and baseline condition (Bottom). Each participant was assigned to only one treatment.

Counterfactual interpolations allowed users to identify the model's main pattern: the position of the *arms* of Stretchy and Sticky. They did this with high certainty, as 83.34% strongly agreed with the corresponding statement. They were more certain about this pattern than with the baseline technique ($H(1) = 8.86$, $p = 0.018$), even though the baseline technique also performed well for this task. The large majority (70%) also identified the color bias with counterfactual interpolations, while only 43% identified this bias using the baseline explanations. However, the difference in rating between conditions for the corresponding statement about color bias was not significant ($H(1) = 3.21$, $p = 0.42$). Participants who had missed the color bias using our method were later asked to provide their reasoning. A participant stated: *"I would think that the color would be relevant if I saw an example where it went from certain to very certain and only the color, brightness or intensity changed."* Such rule-based rather than probabilistic cognitive models of the network may have led users to reject the presence of color bias even though we instructed them clearly that interpolation would only change relevant features.

To our surprise, fewer participants noticed the network's more subtle bias towards object rotation in both conditions. As Figure 4 indicates, participants were somewhat undecided about the relevance, leaning rather to conclude that the network is not sensitive to rotation. As a limitation, we note that participants may not have noticed the rotation bias due to how we had phrased the corresponding statement. When we asked them to explain their reasoning, many explained that they instead focused on the individual blocks' rotation rather than the whole animal.

Both explanation techniques allowed participants to confidently reject statements about irrelevant patterns (sensitivity to the background, sensitivity to the type of blocks). We argue this indicates a high quality of collected responses and good utility of both explanation techniques. Somewhat worrying is participants' assessment of the system's maturity. They were very confident that the network has learned the right patterns and is ready to use for both techniques. Such bias towards model deployment has previously surfaced in other studies (Kaur et al., 2020).

Explanation Satisfaction ratings were very high for both techniques (see Figure 10 in Appendix) underlining that participants perceived both methods very well. While this also means that our method was unable to outperform the baseline, it also shows that our careful visual design and our clear instructions how to use the explanations technique were well received. As a limitation, we note that participants may have found the introductory videos very informative as many reported enjoying the study. This may have led them to more favorable ratings and the conclusion that they understand the system very well regardless of the explanation technique they had used.

## 4 RELATED WORK

Others have suggested methods for counterfactual generation. Chang et al. (2019) identifies relevant regions by optimizing for sufficiency and necessity for the prediction. The classifier is then probed

on the counterfactuals replacing relevant regions with heuristical or generative infilling. Goyal et al. (2019) find regions in a distractor image that would change the prediction if present. Both works assume that relevant features are localized, but for many datasets these may cover the entire image, e.g. changes due to gender or age in face images. Singla et al. (2020); Liu et al. (2019); Baumgartner et al. (2018) explain a black-box neural network by generating counterfactuals with GANs which can generate counterfactuals of similar or even better visual quality. However, the GANs model does not have to align with the explained model perfectly, e.g. see Figure 3 in (Singla et al., 2020).

The TCAV method (Kim et al., 2018) estimates how much manually defined concepts influence the final prediction. Recent work has extended TCAV to discover concepts using super-pixels automatically (Ghorbani et al., 2019). Goyal et al. (2020) extend TCAV to causal effects of concepts and use a VAE as generative model.

Being able to interpolate in feature space and inverting these latent representations is one of the advantages of invertible networks (Jacobsen et al., 2018; Kingma & Dhariwal, 2018). Mackowiak et al. (2020) use invertibility to improve the trustworthiness but focus on out-of-distribution and adversarial examples. (Rombach et al., 2020; Esser et al., 2020) employ invertible networks to understand vanilla convolutional networks better.

One example of an interpretable model is ProtoPNet (Chen et al., 2019). The feature maps of image patches that correspond to prototypical samples in the dataset are used for the final prediction. This way, a result can be explained by pointing to labeled patches. The method is limited to a fixed patch size and does not allow counterfactual reasoning. Another patch-based interpretable model is proposed in Brendel & Bethge (2018).

Our combination of PCA and invertible neural networks for interpretability is novel. The finding that the directional derivative corresponds to ideal counterfactuals, whereas the gradient does not, has not been reported before. We are also not aware of a user study has previously demonstrated that visual counterfactual can help users identify biases of a neural network.

## 5 DISCUSSION

A disadvantage of our method is that it requires an invertible network architecture — the weights of an existing CNN cannot be reused. Learning the input distribution entails additional computational costs, when training an invertible neural network. For non-image domains such as natural language or graphs, the construction of an inverse is currently more difficult. However, first works have taken on the challenge (MacKay et al., 2018; Madhawa et al., 2019). Furthermore, learning the input distribution requires a larger network. Given that our method performed similar to the baseline in the user study in all but one category, an obvious question is whether it is worth the additional effort.

However, the same question applies to almost any explanation method and remains largely unanswered. Unfortunately user evaluations that include a reasonable baselines are very rare. An additional finding of this work is that explanation methods should be evaluated for their *utility and usability* against a *reasonable baseline*. For image classification our work shows, that studying the raw input and corresponding predictions is such a reasonable baseline.It has the potential to allow lay users to identify, many but not all, high level features used for prediction. Even though we found a strong baseline, the user study also demonstrated that our method is useful to lay users as they found two out of three relevant patterns and rejected two more irrelevant patterns. It also highlights that some more subtle patterns may still go unnoticed even when using our method.

We would like to argue that the additonal effort required to implent invertability, may well be justified especially in high-stakes domains. Combining an invertible neural network with a linear classifier enables the use of simple explanation techniques which are otherwise restricted to low complexity models. Here, we can use them on a deep model with much greater predictive power. Counterfactuals can be created by simply using the weight vector of the classifier. In contrast to many other techniques, they are faithful to the model, changing only features relevant for the prediction. Since, they can be inverted back to the input space the high level features they encode are human interpretable. This allows users to discover hypotheses about the models used patterns largely independent of their preconception about feature importance. Using our method we found biases in three datasets including some that have not been previously reported. As we have demonstrated in this work, that *invertibility has mayor advantages for interpretability*.

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

| Dataset | NLL | Accuracy | Supervised |
|---------|-----|----------|------------|
| BLACK MICE | 3.28 | 86.5 | 86.4 (our model) / 88.5±2.5 (Andresen et al., 2020) |
| CELEBA | 2.55 | 88.2 | 89.9 |
| TWO4TWO | 0.63 | 84.9 | 98.2 |

Table 1: Model Performances on the different datasets. Negative Log Likelihood in bits/pixels.

## A APPENDIX: TECHNICAL DETAILS

### A.1 NEURAL NETWORK ARCHITECTURE

Our model follows the Glow model closely (Kingma & Dhariwal, 2018). Similarly, we use a block of *actnorm*, *invertible* $1 \times 1$ *convolution* and *affine coupling* layer. After 18 blocks, we add an reshuffle operation to reduce the spatial dimensions by a factor of 2 and half of the channels are faded out. The first layer is The classification is done before the final mapping to the prior $\mathcal{N}(0, 1)$.

As described in section 2, we trained added the classifier after layer 321 before the final layer 351. Let $\varphi$ denote the first 321 layers and $\mu : \mathbb{R}^n \mapsto \mathbb{R}^n$ the last. We train $\varphi$ both on a supervised loss from the classifier $f(\boldsymbol{x})$ and an unsupervised loss from matching the prior distribution $\mathcal{N}(0, I)$ and the log determinante of the Jacobian. $\mu$ is only trained on the unsupervised loss:

$$\arg \min_{\theta_\varphi, \theta_\mu, \theta_f} L_{\text{un}}(\mu \circ \varphi(\boldsymbol{x})) + \beta \, L_{\text{sup}}(\boldsymbol{w}^T \varphi(\boldsymbol{x}) + b, \, y_{\text{true}}). \tag{1}$$

For the supervised loss $L_{\text{sup}}$, we use the binary cross entropy although our method is not restricted to this loss function and could be extend to more complex losses easily. As unsupervised loss $L_{\text{un}}$, we use the commonly used standard flow loss obtained from the change of variables trick Dinh et al. (2016). The unsupervised loss ensures that inverting the function results in realistic looking images and can also be seen as a regularization.

In total, $\varphi$ and $\mu$ have 158.769.600 parameters. We use the identical network architecture on all datasets.

### A.2 DETAIL TO INTEGRATION: SECTION 3

In section 3.2, we integrated the gradient and the directional dirivative. We used the `torchdiffeq` package. For figure 3b, we integrated from t=[0, 11] using the midpoint method with 20 steps. Here the integration was done in layer 40. As this was rather slow, we used 5steps and t = [0, 4] to determine the differences in the unrelated factors $\boldsymbol{\xi}$ again in 40, shown in Figure 3c.

## B BLACK MICE

In this case study, we apply our method on the BLACK MICE dataset (Andresen et al., 2020). In contrast to CELEBA, the images vary more strongly in location, size, posture, and camera angle. The dataset contains a total of 32576 images of 126 individual mice. Andresen et al. (2020) trained a ResNet and reported an accuracy of 88.5±2.6% using 10-fold cross-validation. Our model achieves a similar accuracy of 86.5% tested on a single fold. The images were collected for earlier works (Hohlbaum et al., 2018; 2017). The mice were divided in three groups: castration, only anesthesia, or untreated. A binary label marks any signs of post-surgical/-anesthetic effects. According to (Langford et al., 2010), typical features for pain are squeezed eyes, pulled-back ears, bulged cheeks and nose, and change in whisker position.

Together with the authors of (Andresen et al., 2020), we reviewed our model's explanations. We confirmed that counterfactuals affect different image features accordingly: eyes, nose, ears, whiskers, and head position change in biologically plausible ways. The mice's eyes seem to be less relevant to the network. For humans, squeezed eyes are a good indicator of pain. However, the counterfactuals only showed slight changes: sometimes the eyes blend into the surroundings. As neural networks perform well on the task using only the eyes (Neurath, 2020), we believe changes to our network architecture could preserve these details. Some other features may co-appear with image artifacts, e.g. the ear's shape changes may also appear partially blended with the background.

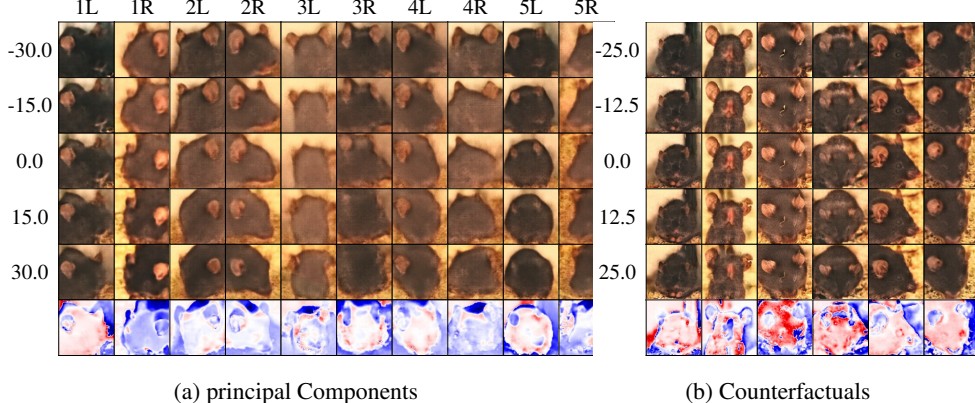

|  (a) principal Components  |  (b) Counterfactuals  |

Figure 5: Examples for the BLACK MICE dataset. **(a)** Random eigenvectors of the isosurface (the columns correspond to the principal components) **(b)** Counterfactuals generated by our model. In both subplots, the rows correspond to the indicated change in logits.

Intriguingly, the counterfactuals also show contrast changes in the surroundings, see figure **??**. The authors of (Andresen et al., 2020) voiced the suspicion that this may be explained how the photos were taken. Since mice after anesthesia or surgery predominantly drop the head and the nose tip points downwards, the camera angle may have been adjusted to get a better view of the animal and, in effect, show more of the yellow wooden bedding material on the cage floor.

To verify if wooden bedding material is predictive, we annotated 1000 randomly selected images from our test set. Depending on the image area covered by the wooden bedding material, we assigned each sample to the classes: $(0) \leq 5\%$, $(1) \leq 20\%$, $(2) > 20\%$ if the bottom of the image showed yellow bedding material. This classification resulted in 346, 258, 396 samples per bin. Of all samples, 44.7% were marked to show post-surgical/-anesthetic effects. Per bin, the label was unevenly distributed: 19.9%, 52.3%, 61.4%. We account for the unequal distribution of labels using partial correlation (see Appendix **??**) and obtain the following values between the models' output probabilities and the bins (95% CI): (1) -0.255 $(-0.31, -0.20)_{95\%}$, (2) 0.026 $(-0.04, 0.09)_{95\%}$, (3) 0.217 $(0.16, 0.27)_{95\%}$.

The label "post-surgical/-anesthetic effects" is unequally distributed across the three bins: 346, 258, 396. This can be problematic, when we measure the correlation between sample's bin and the model logit score. The model has learned to predict lower scores for a negative label and vise versa. To account for this, we calculate the partial correlation between the model's output probability and the bin class while using the label as a confounding variable. In table 2, we report both full and partial correlations and also the correlations of the bins with the label.

These results confirm a connection between the surroundings, label, and logit score. The hints of our explanations to this bias in the data were correct. The surroundings' changes can be explained probably by mice dropping their head if in pain and by changes to the camera angle. As we could also confirm many characteristic features, the network does not base its decision solely on wooden bedding material. This case study highlights the practicability of our method in a real-world scenario.

Table 2: Bias in the BLACK MICE dataset. The post-surgical/-anesthetic effects label is unevenly distributed across the bins (0-2) for the amount of yellow bedding material present in an image. The classifier's probabilties are correlated negatively with (0) fewer bedding material and positively with more (2). When we account for the effect of unequal label distribution using partial correlation, the output probabilities and bins (0 & 2) are still correlated.

| Bin | Data | Method | Corr. | CI 95% |
|---|---|---|---|---|
| (0) $\leq 5$ | Label | full | -0.362 | -0.410, -0.310 |
| (1) $\leq 20$ | Label | full | 0.090 | 0.030, 0.150 |
| (2) $> 20$ | Label | full | 0.271 | 0.210, 0.330 |
| (0) $\leq 5$ | Pred. Prob. | full | -0.429 | -0.480, -0.380 |
| (1) $\leq 20$ | Pred. Prob. | full | 0.085 | 0.020, 0.150 |
| (2) $> 20$ | Pred. Prob. | full | 0.341 | 0.290, 0.390 |
| (1) $\leq 5$ | Pred. Prob. | partial | -0.255 | -0.310, -0.200 |
| (2) $\leq 20$ | Pred. Prob. | partial | 0.026 | -0.040, 0.090 |
| (3) $> 20$ | Pred. Prob. | partial | 0.217 | 0.160, 0.270 |

## C TWO4TWO

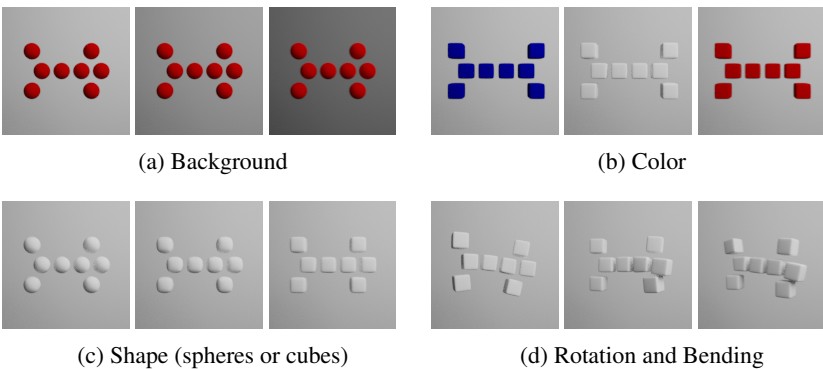

(a) Background

(b) Color

(c) Shape (spheres or cubes)

(d) Rotation and Bending

Figure 6: Parameters in the TWO4TWO dataset. The objects in the TWO4TWO dataset are *Sticky* shown in (a) and *Stretchy* shown in (b). Each animal consists of a spine of four blocks and two sets of arms at either end. For *Sticky*, the right set of arms is moved inwards. For the *stretchy* class, both sets of arms are moved outwards. (a) background and (b) animal colors can be changed. (c) The individual blocks can be spherical, cubic or something in between This is achieved by rounding off cubes until they become spherical. (d) The animals can take a random pose.

| Parameter | Corr. Data | CI95% | Corr. Change | CI 95% | |
|---|---|---|---|---|---|
| Color | 0.329 | 0.27, 0.38 | 0.352 | 0.33, 0.37 | |
| Background | 0.022 | -0.04, 0.08 | -0.037 | -0.06, -0.01 | |
| Incline | 0.032 | -0.03, 0.09 | 0.003 | -0.02, 0.02 | |
| Arm Position | -0.799 | -0.82, -0.78 | -0.798 | -0.81, -0.79 | |
| Spherical | -0.053 | -0.11, 0.01 | -0.010 | -0.03, 0.01 | |
| Abs. Rotation | 0.060 | -0.00, 0.12 | 0.136 | 0.11, 0.16 | |

Table 3: TWO4TWO: Correlation between object parameters and the model's output probabilities. *Corr. Data*: Correlation estimated on the joint distribution. *Corr. Change*: Correlation if only the parameter is changed and all other parameters are kept fixed. While the absolute rotation is only slightly correlated with the model output when calculating correlation on the test set, it becomes correlated if we solely change the attribute and keeping all others fixed.

## D CELEBA

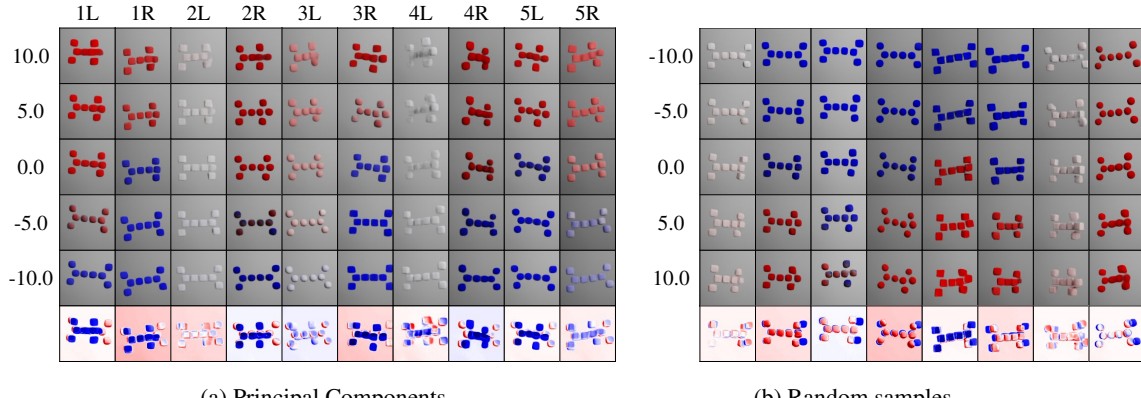

(a) Principal Components        (b) Random samples

Figure 7: Examples for the TWO4TWO dataset. **(a)** Random eigenvectors of the isosurface (the columns correspond to the principal components) **(b)** Counterfactuals generated by our model. In both subplots, the rows correspond to the indicated change in logits.

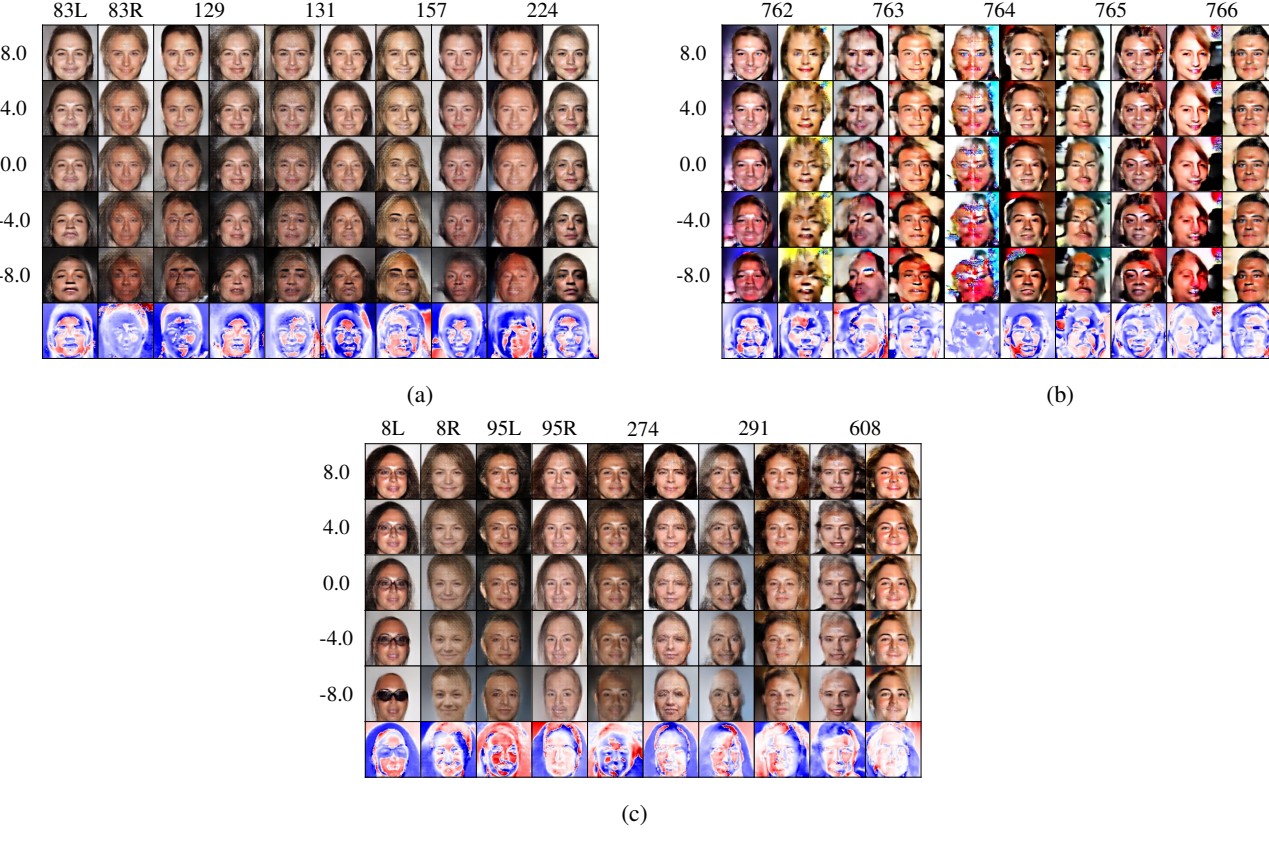

Figure 8: Eigenvectors of the isosurface for the "attractiveness" class. **(a)** Eigenvectors with the higest explained variance **(b)** Randomly selected eigenvectors: 82, 129, 131, 157, 224 **(c)** Eigenvectors with the smallest explained variance

# E   USER-STUDY PREGISTRATION AND HYPOTHESIS

The Study is preregisterd at `https://aspredicted.org/` we provide and anonimyzed pdf version in supplemental material. Participants (N=60), of the study were required to fluent in English and needed to have an approval rate of at least 95. Given the demanding nature of the task and

| Name | Attr. % | freq. | r | r Gender | Name | Attr. % | freq. | r | r Gender |
|---|---|---|---|---|---|---|---|---|---|
| **Beard** | 59.13 | | 0.36 | | Attractive | 81.20 | 49.58 | 0.63 | 0.53 |
| No Beard | 59.03 | 85.37 | 0.35 | 0.06 | Bags Under Eyes | 31.79 | 20.26 | -0.24 | -0.07 |
| Five o Clock Shadow | 28.23 | 9.99 | -0.15 | 0.14 | Bangs | 64.65 | 15.57 | 0.11 | -0.00 |
| Goatee | 7.43 | 4.58 | -0.24 | -0.09 | Big Lips | 63.20 | 32.70 | 0.15 | 0.06 |
| Mustache | 6.61 | 3.87 | -0.22 | -0.09 | Big Nose | 31.03 | 21.20 | -0.28 | -0.12 |
| Sideburns | 11.23 | 4.64 | -0.21 | -0.07 | Black Hair | 51.59 | 27.16 | -0.00 | 0.07 |
| **Makeup** | 80.63 | | 0.68 | | Blond Hair | 73.27 | 13.33 | 0.17 | 0.02 |
| Wearing Lipstick | 80.64 | 52.19 | 0.64 | 0.35 | Blurry | 21.39 | 5.06 | -0.14 | -0.17 |
| Heavy Makeup | 86.08 | 40.50 | 0.62 | 0.38 | Brown Hair | 68.97 | 17.97 | 0.19 | 0.15 |
| Rosy Cheeks | 90.92 | 7.17 | 0.26 | 0.16 | Bushy Eyebrows | 54.25 | 12.95 | 0.03 | 0.22 |
| Arched Eyebrows | 79.87 | 28.44 | 0.38 | 0.19 | Eyeglasses | 3.49 | 6.46 | -0.35 | -0.29 |
| **Hairloss** | 56.30 | | 0.30 | | Gray Hair | 2.20 | 3.19 | -0.25 | -0.19 |
| Bald | 0.71 | 2.12 | -0.22 | -0.14 | Male | 20.03 | 38.65 | -0.59 | |
| Receding Hairline | 23.85 | 8.49 | -0.22 | -0.19 | Narrow Eyes | 45.22 | 14.87 | -0.06 | -0.09 |
| **Cubby/Double Chin** | 56.21 | | 0.32 | | Oval Face | 62.96 | 29.56 | 0.19 | 0.16 |
| Chubby | 8.22 | 5.30 | -0.29 | -0.20 | Pale Skin | 80.12 | 4.21 | 0.12 | 0.10 |
| Double Chin | 6.35 | 4.57 | -0.26 | -0.17 | Pointy Nose | 70.95 | 28.57 | 0.27 | 0.18 |
| **Smiling/Mouth/Cheek** | 62.11 | | 0.25 | | Straight Hair | 53.89 | 20.99 | 0.01 | 0.08 |
| High Cheekbones | 63.21 | 48.18 | 0.23 | 0.08 | Wavy Hair | 70.85 | 36.40 | 0.33 | 0.17 |
| Mouth Slightly Open | 53.85 | 49.51 | 0.02 | -0.05 | Wearing Earrings | 70.93 | 20.66 | 0.20 | -0.01 |
| Smiling | 60.54 | 50.03 | 0.19 | 0.12 | Wearing Hat | 12.99 | 4.20 | -0.20 | -0.15 |
| | | | | | Wearing Necklace | 72.61 | 13.79 | 0.18 | 0.02 |
| | | | | | Wearing Necktie | 13.65 | 7.01 | -0.27 | -0.08 |
| | | | | | Young | 62.63 | 75.71 | 0.44 | 0.37 |

Table 4:

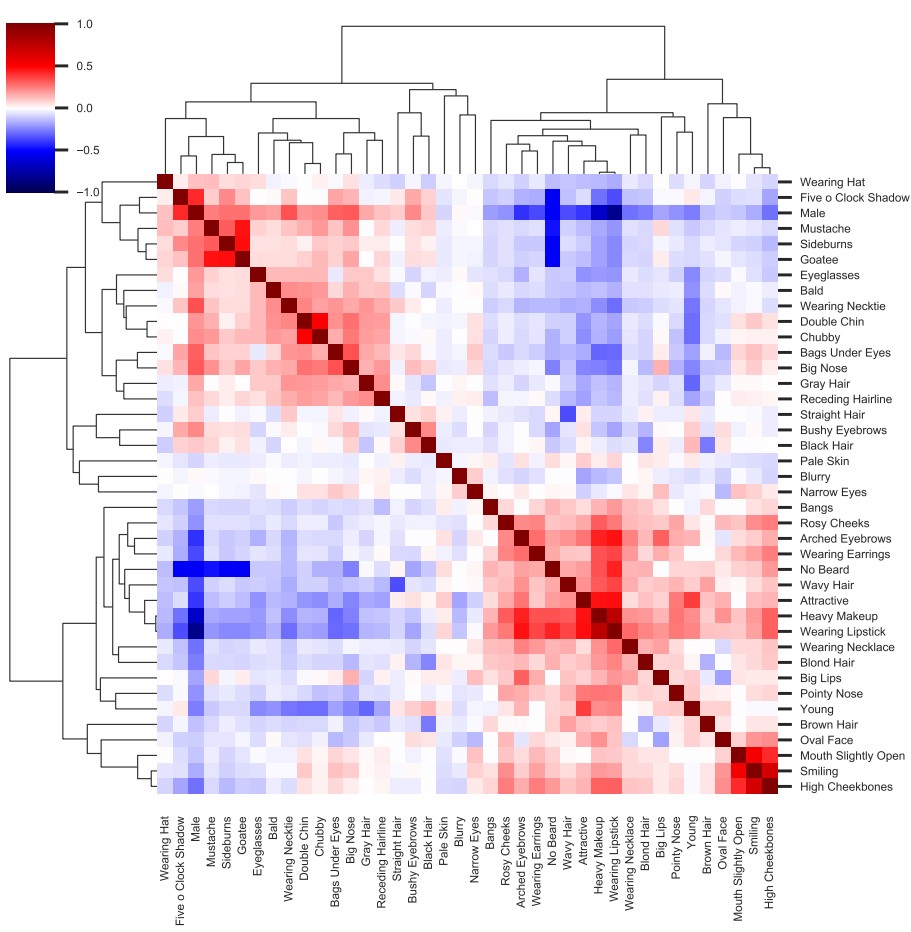

Figure 9: CelebA correlation matrix indicating the relationship among the annotated labels. The labels are sorted according to a hierarchical clustering on the correlation values. There are two strong clusters of labels (upper left and lower right), in which e.g., the label *Attractiveness* belongs to the same cluster (lower right) as *Wearing Lipstick* while the label *Male* belongs to the other cluster. This confirms the biases highlighted by our method.

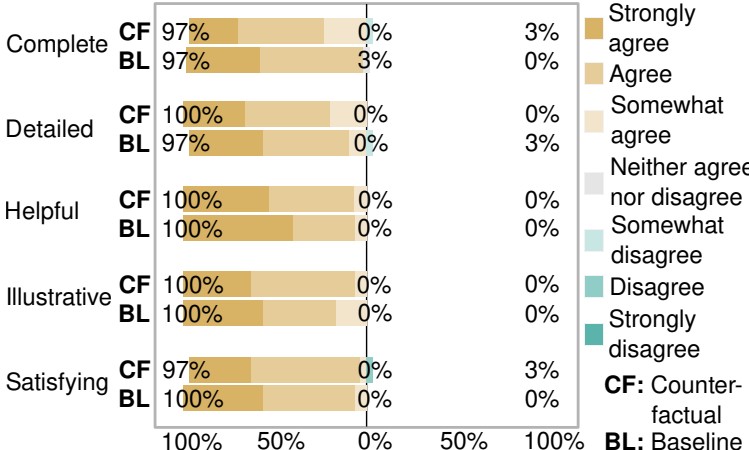

Figure 10: Explanation Satisfaction Ratings by our study participants for counterfactual interpolations (CF) and example-based explanation baseline (BL)

the complex of concepts used in the instructions they also needed to have an academic degree in Computer Science, Engineering, Finance, Mathematics, Medicine, Physics or Psychology. Figure 10 summarizes the subjective ratings participants gave about the two explanation technqiues used in the study.

### E.1 EVALUATION OF PREREGISTERED HYPOTHESIS

H1: The hypothesis *"Studying the system's predictions on the validation set (Baseline Explanation technique - referred to as Baseline) allows users to verify that the neural network (NN) is using the blocks spatial arrangement (Pattern 1) for its predictions of the abstract animals."* is confirmed as 83.33% at least somewhat agree to corresponding statement.

H2: The hypothesis *"Baseline does not allow users to detect the NN bias for colour (Pattern 2) and rotation (Pattern 3)."* is confirmed. Only 46.66 % of users at least somewhat disagree with the statement claiming that there is a rotation pattern while only 43.33% at least somewhat agree (the remaining are undecided). For the colour pattern 50% at least somewhat disagree that there is such a pattern and only 43.33% at leat somewhat agree.

H3: Duplicate of H2 (copy and paste error during preregistration)

H4: The hypothesis *"Studying the system's predictions with counterfactual interpolations as explanations (referred to as NNWI) allows users to verify that NN is using Pattern 1."* is confirmed as 96.66% at least agree with the corresponding statement.

H5: The hypothesis *"Counterfactual interpolations allows users to detect Pattern 2 and Pattern 3."* is rejected. While 70 % at least somewhat agree with the statement about Pattern 2 only 33.33% at least somewhat agree with the statement about Pattern 3.

H6: The hypothesis *"Counterfactual interpolations allows users to verify that NN is neither using the background of the image (Pattern 4) nor the surface structure of objects (Pattern 5)."* is confirmed. The corresponding statement about Pattern 4 and 5 have been at least somewhat disagreed to by 83.33% and 73.33% respectively.

H7: The hypothesis *"Counterfactual interpolations allow users to detect Pattern 1 with higher confidence"* is confirmed. Agreement with the corresponding statement was significantly different between conditions (p = 0.003) and on average higher for Counterfactual Interpolations (2.67) compared to the baseline (1.76).

H8: The hypothesis *"Counterfactual interpolations allow users to reject Pattern 4 and Pattern 5 with higher confidence"* is rejected. There was no significant difference in the certainty for disagreeing with corresponding statements.

H9: The hypothesis *"Counterfactual interpolations allow users to detect Pattern 2 and Pattern 3 with higher confidence."* is rejected since H5 was rejected. However, it is worth pointing out that for the statement about color 70% of participants at least somewhat agreed to it if they received counterfactual interpolation while only 43.33% at least somewhat agreed to it if they received example based explanations.

H10: The hypothesis *"Counterfactual interpolations leads users to be more confident in the matureness of the system."* is rejected as agreement to the corresponding statement was not significantly different across conditions. In both conditions participants where rather confident in the system.

H11: The hypothesis *"Users are more satisfied with Counterfactual interpolations as explanations."* is rejected. Explanation Satisfaction rating were very high in both conditions but not significantly different.

## F    IMAGE SOURCE

As the copyright of CELEBA is unclear and includes images under no free license, we decided against showing any original CELEBA images in the paper. We show the following these six images all under permissive license:  Obama (CC BY 3.0): `https://commons.wikimedia.org/wiki/File:Official_portrait_of_Barack_Obama.jpg`

Commander, Eileen M. Collins (Public Domain): `https://www.flickr.com/photos/nasacommons/16504233985/`

Carl Jacobi (Public Domain): `https://de.wikipedia.org/wiki/Datei:Carl_Jacobi.jpg`

Grace Hopper (Public domain): `https://de.wikipedia.org/wiki/Datei:Grace_Hopper.jpg`

Alan Touring (CC BY 4.0): `https://commons.wikimedia.org/wiki/File:%D0%A2%D1%8C%D1%8E%D1%80%D0%B8%D0%BD%D0%B3.jpg`

Lyndsey Scott (CC BY 4.0): `https://en.wikipedia.org/wiki/File:Lyndsey_Scott_being_combed.jpg`

