# OpenReview forum: "Interpretability Through Invertibility: A Deep Convolutional Network With Ideal Counterfactuals And Isosurfaces"
_ICLR.cc/2021/Conference — Reject_

### Official Review · AnonReviewer2 · 2020-10-27
**Interesting idea needing more work**

**Rating:** 6
**Confidence:** 4

**Review:**

Update after revision
------------------------------
I thank the authors for their work on this paper. The second reading was more pleasant. I agree with the authors that performing a user-study is an important effort, that should be encouraged. I however still believe that, if not benefitial to the user, the complexity of the method can be a drawback. I also wished that more comparisons, but especially other data modalities were investigated. I have updated my rating to reflect the improvement in the text.

Short summary
-----------------------
The authors propose a technique based on an invertible network to provide counterfactuals relative to one class of interest. The counterfactuals can be interpolated across an isosurface, displaying parameters which do not affect the model’s decision. The authors propose an attribution map based on those counterfactuals and evaluate counterfactuals in a qualitative manner, based on their own observations on 3 datasets, as well as based on a human-grounded evaluation on a synthetic dataset.

Strengths
---------------
The use of an invertible dataset is rather novel in the field of explainability, and the relationship between the obtained counterfactuals and gradient-based interpolation methods is interesting. The human-grounded evaluation is definitely a large undertaking that is not often performed to assess the usefulness of interpretability techniques.

Weaknesses
-------------------
I have identified several weaknesses of the work that justify my recommendation:
- the (lack of) clarity of the text.
- the assessment of the technique, as the results of the human-grounded evaluation are mixed, with users not being significantly more accurate in finding confounding factors compared to a baseline technique.
- the limitations of the technique, not discussed in depth. For instance, I can see difficulties in evaluating the effect of classes that are not present as “training classes” in the dataset, which requires a large labeling effort. In addition, how the technique would transpose to non-image datasets, or whether there are limitations in the invertible architectures to consider should be mentioned.

Novelty
-----------
The “Related works” section is rather limited, which makes it difficult to evaluate. In general, the use of invertible networks as interpretable networks is novel.

Clarity
---------
Clarity was a major weakness of this work for me:
- the datasets are illustrated in figures but not mentioned until much later
-  the maths are described in sections that seem unrelated to each other, without depicting the relationships between the different steps
- multiple concepts are unclear (see detailed comments)
- the motivations are not clearly explained

Rigor
--------
I found the qualitative evaluation on the 3 datasets unconvincing, as it is unclear whether the same conclusions could not have been reached using other techniques.
While I was most interested by the discussion around the generation of counterfactuals based on the invertible network compared to based on the integration of gradients, I wished there was a definition of an “ideal” counterfactual, qualitative or (preferably) quantitative. The single example provided in the main text is appealing but this requires more evidence to me.
Finally, the “saliency” maps defined in this work do not seem to be used later on in the work. I doubt that looking at them would improve human evaluation of a model’s behavior.

Detailed comments
-----------------------------
- Counterfactuals: their quality seems subject to appreciation and confirmation bias, especially on potentially cherry picked examples. To assess their quality, I would suggest to use the BAM dataset (Yang and Kim, 2019, https://github.com/google-research-datasets/bam) which was generated to benchmark attribution methods. I would overall suggest the use of this dataset for assessing the faithfulness (sensitivity, specificity) of the proposed approach.
- The choice of the mice dataset should be justified as this doesn’t seem like an obvious choice to assess the quality of attribution techniques. It is quite difficult to estimate any effect, and feels like qualitative evaluation is biased by the authors’ remarks given the lack of knowledge of the problem.
- There should be more details about the Two4Two dataset and its motivations, as well as how it relates to other datasets (e.g. Goyal et al., 2019)
- How does the proposed approach relate to “completeness” (Sundararajan et al., 2017)?
- What is the mathematical justification to resize the saliency map of an intermediate layer to the input resolution? Is there a citation for this process showing that this is a reasonable assumption?
- I am confused by the section on saliency maps: what does h represent? The activations at an intermediate layer? The motivation is unclear: what are the authors trying to highlight in these “saliency maps”? Are these computed attributions or are these L1 distance between activations (in %) between x and x_tilde? Or is it a cosine distance (as suggested by the next sentence mentioning the angle?)
- The tasks used for illustration are not described in the text. Examples of y and epsilon should be provided.
- Is the technique limited to the model’s predicted classes?
- How is “ideal” counterfactual described and mathematically verified?
- The relationship between counterfactuals and e.g. integrated gradients is unclear: the first clearly needs a model that can generate data, while the latter integrates the gradients between a baseline (defined by the user) and the input. More details and explanations are required to make this relationship clearer.
- What are the participants in the human-based study viewing? Are they comparing the counterfactuals to e.g. SmoothGrad maps, or the saliency as defined per the proposed approach?
- It is unclear what the participants answered: Figure 5a mentions that the main score is “strongly disagree” for “arms” (both baseline and interpolation) while the text refers to “strongly agree”. Example questions would help.
- The results of the human-grounded study are not very conclusive. Note: please correct for multiple comparisons due to multiple statistical testing of the same effect.
- Kim et al., 2018 already displayed that human users were performing poorly at identifying a network’s decision behavior based on saliency maps. A better comparison could have relied on TCAV instead, especially as the concepts can easily be mapped to the features given the synthetic dataset. This could have made a stronger case for the use of invertible networks, especially as Goyash et al (2019) mention the use of counterfactuals based on concepts.
- How about non-image datasets?

Minor
-------
- Intro: I would suggest using “transparency” rather than “interpretability” when referring to logistic regression (e.g. Lipton, 2016). The interpretability of linear model weights is indeed debatable, as weights will depend on the regularization and signal-to-noise ratio in the data (Haufe et al., 2014).
- No clear flow between the different works in the intro. No clear motivation behind counterfactuals.
- proofreading: paper is quite hard to follow and minor changes to grammar (e.g. “Their similarity is easy to seen”) makes it more difficult to assess. The quality of the writing deteriorates in sections 3, 4 and 5.
- It is unclear what scale delta epsilon represents, and whether we can expect the norm of the different techniques to be comparable.

---

> ### Author Response · Authors · 2020-11-25
> **R2**
>
> R2: Interesting idea needing more work
> > the (lack of) clarity of the text.
> > the assessment of the technique, as the results of the human-grounded evaluation are mixed, with users not being significantly more accurate in finding confounding factors compared to a baseline technique.
>
> This is correct. However, please also take into your consideration that the study still demonstrates the usefulness of our method as well as some of its shortcomings. In contrast to many other evaluations, which do not even consider baselines (e.g. Ribeiro et al. (2016); Singla et al.(2020)) or don’t even evaluate their methods with human subjects at all, we made an effort to create a simple but strong baseline, taking into consideration findings from HCI about usability issues of explanation techniques.
>
> We would like to ask reviewers to consider the rigor put into the study design and the identification of a good baseline technique as additional contributions of our paper. After all, this might inspire more human evaluations in the machine learning community, if they are deemed valuable.
>
> > the limitations of the technique, [...] should be mentioned.
>
> We mentioned the added computation costs in our submitted version and also the challenge and possible approaches to apply invertible networks to RNN or GraphNN. We now also state clearer that our method requires us to use a custom network architecture. Since labeling data isn’t a requirement specific to our model, but affects rather all architectures we have not mentioned this point.
>
> > Novelty, The “Related works” is rather limited,
>
> We agree and rewrote the related work section and commented on the similarities and differences in greater detail.
>
>
> > [Rigor] I found the qualitative evaluation on the 3 datasets unconvincing [...]
>
> We agree that possible conclusions might have reached using other techniques. In our user-study, we show that users can reach similar conclusions using a simple baseline. We now state in the related work section that GAN based counterfactuals could probably create similar looking images. However, they cannot guarantee that the explanations are faithful to the model.
>
> > [Rigor] While I was most interested by the discussion around the generation of counterfactuals based on the invertible network compared to based on the integration of gradients, I wished there was a definition of an “ideal” counterfactual, qualitative or (preferably) quantitative. The single example provided in the main text is appealing but this requires more evidence to me.
> We agree and have now included such a definition at the beginning of section 2.
>
> We agree that saliency maps can be inconclusive, no matter what method has been used to generate them. There are more and more studies being published showing evidence for that. We have now included a small summary of such findings in our paper along with the remark that the same useability concern applies to our saliency maps as well. Regardless, our saliency maps are faithful to the model, which is an important and unique contribution in comparison to other methods. A saliency map of a counterfactual highlighting the entire face is still correct as many labels (gender, ethnicity, age, attractiveness) impact the whole face.
>
> > Counterfactuals: their quality seems subject to appreciation and confirmation bias, especially on potentially cherry picked examples.
>
> The quality of counterfactuals is evaluated in the users study, where they were picked randomly not manually. Most examples in the paper are based on the principal components of the dataset (see Figure 1b) and therefore not cherry picked. We also show a number of random samples along principal components in the Appendix.
>
>
> > There should be more details about the Two4Two dataset and its motivations, as well as how it relates to other datasets (e.g. Goyal et al., 2019)
>
> We provide an additional description of the Two4Two dataset in the appendix and relate it to other datasets, e.g. the BAM dataset.
>
> > How does the proposed approach relate to “completeness” (Sundararajan et al., 2017)?
>
> Completeness requires that the sum of the attribution map equals the difference of the logit score between the image and baseline. Our saliency maps do not fulfill completeness as defined by (Sundararajan et al., 2017). We do look at the differences between counterfactual examples and they can be arbitrarily large as φ can stretch the space. As a side note: we believe that “Completeness” fails to account for the non-linear stretching of neural networks and is not a property to require (but this is a different discussion).
>
> > What is the mathematical justification to resize the saliency map [...]
> As activations remain localized in a convolutional network, this operation is valid. The lower resolutional feature map still matches the feature locations well. Grad-CAM for example does the rescaling even using the last convolutional feature map.

---

> > ### Author Response · Authors · 2020-11-25
> > **R2**
> >
> >
> > >I am confused by the section on saliency maps: what does h represent? The activations at an intermediate layer? The motivation is unclear: what are the authors trying to highlight in these “saliency maps”? Are these computed attributions or are these L1 distance between activations (in %) between x and x_tilde? Or is it a cosine distance (as suggested by the next sentence mentioning the angle?)
> >
> > We agree that the mathematical formulation was not clear. The score is based on the dot-product between the change $|\Delta h|$ and feature activation $h$. We have clarified this in the manuscript.
> >
> > > The tasks used for illustration are not described in the text. Examples of y and epsilon should be provided.
> >
> > We added a respective comment to the manuscript.
> >
> >
> > > Is the technique limited to the model’s predicted classes?
> >
> > You could add classifiers, finetune them and then explain them. Or you could also cluster intermediate features and invert them back.
> >
> > > How is “ideal” counterfactual described and mathematically verified?
> >
> > We added a definition of ideal counterfactual. Mathematically, there must exist a path from a startpoint $x$ to $\tilde x$ such that the gradient of the path is perfectly aligned with the gradient of the classifier.
> >
> > > The relationship between counterfactuals and e.g. integrated gradients is unclear: [..]
> > We have improved the description of this section substantially.
> >
> >
> > > What are the participants in the human-based study viewing? Are they comparing the counterfactuals to e.g. SmoothGrad maps, or the saliency as defined per the proposed approach?
> >
> > The participants were assigned two groups. Each group only saw one explanation technique (our counterfactuals or the baseline). Hence, they are not comparing methods. We reworked the section to amke this clearer.
> >
> > > It is unclear what the participants answered: Figure 5a mentions that the main score is “strongly disagree” for “arms” (both baseline and interpolation) while the text refers to “strongly agree”. Example questions would help.
> >
> > Unfortunately, the labels were flipped. We apologize and thank you for pointing this out.
> >
> > > The results of the human-grounded study are not very conclusive. Note: please correct for multiple comparisons due to multiple statistical testing of the same effect.
> >
> > We have added a discussion of our results in the manuscript. Please see the general response to points raised by all reviewers.
> >
> > Following your advice, we have now applied the Bonferroni-correction. The results remain unchanged. Thank you for making us aware of this!
> >
> >
> > > Kim et al., 2018 already displayed that human users were performing poorly at identifying a network’s decision behavior based on saliency maps. A better comparison could have relied on TCAV instead, especially as the concepts can easily be mapped to the features given the synthetic dataset. This could have made a stronger case for the use of invertible networks, especially as Goyash et al (2019) mention the use of counterfactuals based on concepts.
> >
> > While it is true that (Kim et al., 2018) show the limitations of saliency maps, our comparison between directional derivative and gradient adds theoretical evidence against gradient-based attribution method.
> >
> > We decided against using TCAV in our evaluation as we tackle different questions. TCAV requires manually labeled concepts and we provide a method to discover possible concepts worth annotating. Instead of reporting the correlations between the different attributes and the logit score, we could have reported the TCAV scores. To us, the simple correlation seems more straightforward and comprehensible.
> >
> > (Goyal et al., 2019) extends TCAV to estimate the causal effect of concepts. We do report a similar causal effect score as Goyal for the Two4Two dataset, where we control the data generation process. The main advantage of our model is that it uses the same model to classify and generate the explanations. While we could have implemented the work by Goyal for invertible neural networks, the potential insights would be a comparison between VAEs and invertible neural networks.

---

> > > ### Author Response · Authors · 2020-11-25
> > > **R2**
> > >
> > >
> > > > How about non-image datasets?
> > >
> > > We discuss other data domains in the conclusion. Our method mainly depends on the availability of an inverse, PCA and the architecture of the network (e.g. where to put the classifier) and could probably be adapted to other domains.
> > > Minor
> > > Intro: I would suggest using “transparency” rather than “interpretability” when referring to logistic regression (e.g. Lipton, 2016). The interpretability of linear model weights is indeed debatable, as weights will depend on the regularization and signal-to-noise ratio in the data (Haufe et al., 2014).
> > > No clear flow between the different works in the intro. No clear motivation behind counterfactuals.
> > > proofreading: paper is quite hard to follow and minor changes to grammar (e.g. “Their similarity is easy to seen”) makes it more difficult to assess. The quality of the writing deteriorates in sections 3, 4 and 5.
> > > It is unclear what scale delta epsilon represents, and whether we can expect the norm of the different techniques to be comparable.

---

### Official Review · AnonReviewer3 · 2020-10-28
**interesting idea but the execution and writing left a lot to be desired, seems not proof-read!**

**Rating:** 5
**Confidence:** 5

**Review:**

Summary:  The paper presents a promising idea to build interpretable models by combining discriminative and generative approach. The proposed model uses an invertible neural network to model the data distribution. The invertibility helps in transforming the learned feature vector back to the image domain. A linear discriminative classifier is trained on the feature vector to perform binary classification. Using the inverse function, the model generates a counterfactual explanation by inverting a modified logit score to create a new image as an explanation. The authors further construct an orthogonal basis using PCA, such that modifying feature vector in those directions results in no change in the classifier's prediction. Decomposing the feature space into such a basis helps discover potential biases in the dataset and the classification model. The experiments compare the proposed method's performance with fully discriminative models and post-hoc interpretability methods such as gradient-based saliency maps.

Major
-----------------

•	The paper an interesting and potentially important idea. But at times, the text is difficult to read.
•	The conclusion from Figure 2 is not clear. From the caption and the figure, it is not clear which attributes such as smiling; gender are important for the classifier's positive /negative attractive decision.
•	In Figure 2 and Figure 3(b), a label showing the different principal components considered in each column, and the logit/prediction of the classifier for each row will improve the figure's readability.
•	Saliency maps highlight important regions of an image for the prediction decision. The saliency maps in Figure 3(a) highlight almost the entire face; hence they are inconclusive.
•	Figure 4(a) shows the final results after integrating the original image along with different derivatives. To understand the results, it would be helpful to show some examples over which the integration took place.
•	An example of a random sample, with minimal/no changes along the independent factors for the proposed method, as compared to positive changes by other methods, will help in understanding the results in Figure 4(b).
•	The number reported at the end of section 3 on "the gradient of x and the directional derivative dx/dw" should be reported in a table to allow a proper comparison between different methods.
•	The directional derivative dx/dw, in the model, is w.r.t the weight vector of the binary classifier, trained to identify the label. Related work by Kim et al. (2018), "Interpretability beyond feature attribution: Quantitative testing with concept activation vectors", also used directional derivatives w.r.t to binary classifiers, trained to identify a human-defined concept. A comparison with this method will help the reader understand the different applications of directional derivatives and how directional derivatives can be used without an invertible network.
•	Table 1 doesn't report the results for the supervised method for celebA and tow4two datasets.
•	The numbers reported in Table 2 lacks a coherent conclusion. The corr. data and corr. change columns have values in similar ranges. Please elaborate and discuss the results.
•	In Figure 5(b), it's not clear how real images for baseline condition are sampled. To allow proper comparison, as in counterfactual design, three images should be selected as primary images, the rest images should be sampled based on its minimum distance to the primary images but a different label.
•	In the Evaluation section, for celebA, the authors identified principal components that correspond to attributes like gender, smiling. The results are reported in a qualitative manner, with inferences draw by showing a few examples only. A quantitative analysis is required to demonstrate the dependency of the "attractive" attribute on other attributes. The results shown in Figure 6 of the appendix don't have a thorough caption to illustrate the findings.

Minor:
------------

•	The caption for Figure 1 has typos.
•	In Figure 4(a), the caption doesn't describe the top label.
•	The opening quotation marks are inverted throughout the text.
•	Table 1 and Table 2 are shown after the references. They should be placed with the main text before the references.
•	The label of Figure 5(b) has a very small font.

---

> ### Author Response · Authors · 2020-11-25
> **R3**
>
> R3: interesting idea but the execution and writing left a lot to be desired, seems not proof-read!
> >  The paper an interesting and potentially important idea. But at times, the text is difficult to read.
>
> We agree with your criticism and reworked the entire manuscript substantially.
>
> > The conclusion from Figure 2 is not clear. [...]
>
> We removed these remarks from the figure caption. Instead, we discuss them in detail in the CelebA evaluation.
>
>
> > In Figure 2 and Figure 3(b), a label showing the different principal components considered in each column, and the logit/prediction of the classifier for each row will improve the figure's readability.
>
> We have adopted your advice, thank you!
>
> > The saliency maps in Figure 3(a) highlight almost the entire face; hence they are inconclusive.
>
> We agree that saliency maps can be inconclusive, no matter what method has been used to generate them. There are more and more studies being published showing evidence for that. We have now included a small summary of such findings in our paper along with the remark that the same useability concern applies to our saliency maps as well. Regardless, our saliency maps are faithful to the model, which is an important and unique contribution in comparison to other methods. A saliency map of a counterfactual highlighting the entire face is still correct as many labels (gender, ethnicity, age, attractiveness) impact the whole face.
>
>
> > Figure 4(a) shows the final results [...] To understand the results, it would be helpful to show some examples over which the integration took place.
>
> Figure 4a (now 3b) shows the original image on the left. We have not included intermediate integration steps as they basically show interpolations between the original and the final image. We, however, provide our code so an interested reader can investigate these steps.
>
>
> > An example of a random sample, with minimal/no changes along the independent factors for the proposed method, [...]
>
> We provide examples with no change along the independent factors are given by the counterfactual interpolation. The astronaut integrated along the directional derivative in the old Figure 4a is an example with minimal change along the independent factors.
>
>
> > The number reported at the end of section 3 on "the gradient of x and the directional derivative dx/dw" should be reported in a table [...]
>
> You are right, we should have reported those numbers in a table. We, however, decided to exclude the self-similarity comparison between the gradient and the directional derivative to focus on the other results.
>
> > The directional derivative dx/dw, in the model, is w.r.t the weight vector of the binary classifier, trained to identify the label. Related work by Kim et al. (2018)[...]also used directional derivatives w.r.t to binary classifiers, trained to identify a human-defined concept. A comparison with this method will help the reader understand the different applications of directional derivatives and how directional derivatives can be used without an invertible network.
>
> We have added a short comment on the relationship with (and the main difference to) TCAV - which does not compute the derivative on an invertible neural network.
>
>
>
> > Table 1 doesn't report the results for the supervised method for celebA and tow4two datasets.
>
> We have now included these numbers in the updated version of the manuscript.
>
> > The numbers reported in Table 2 lacks a coherent conclusion. The corr. data and corr. change columns have values in similar ranges. Please elaborate and discuss the results.
>
> We extended the figures caption to discuss the results.

---

> > ### Author Response · Authors · 2020-11-25
> > **R3**
> >
> > > In Figure 5(b), it's not clear how real images for baseline condition are sampled. To allow proper comparison, as in counterfactual design, three images should be selected as primary images, the rest images should be sampled based on its minimum distance to the primary images but a different label.
> >
> > We have received several comments about the baseline from other reviewers too. We improved the description and motivation of the baseline integrating many of the reviewer’s remarks. Thank you!
> >
> > We understand your suggestion as follows: You suggest to recreate counterfactual interpolation by grouping similar images together for one row but with different logit scores. While this would provide a counterfactual interpolation based on real images, it would likely be disadvantageous for the baseline. Consider that the main pattern in the dataset was that some objects (arms) change their location relative to other objects.  If we measure the similarity in pixel space, similarly colored images would be grouped together -- providing a false impression that color does not change and hence is not important for the prediction. Participants may not notice the colour bias this way. A more complex approach based on intermediate features could come with its own challenges and so we decided for the baseline with least bias.
> >
> > >[...]the authors identified principal components that correspond to attributes like gender, smiling.[...]A quantitative analysis is required to demonstrate the dependency of the "attractive" attribute on other attributes. [...]
> >
> > Following your comment, we included the correlation coefficients between the “Attractive” logit and the other properties in the CelebA dataset.

---

### Official Review · AnonReviewer4 · 2020-10-29
**Use of invertible CNNs to construct counterfactuals and isosurfaces**

**Rating:** 5
**Confidence:** 3

**Review:**

This paper describes a computational method to construct ideal counterfactuals and isosurfaces via invertible CNNs, and uses it to reveal biases in three different datasets.

Strengths:
1. The use of directional derivative to construct ideal counterfactuals is interesting.
2. Leveraging PCA to construct isosurfaces is neat.
3. The human study is a plus, where the stimuli are based on counterfactual interpolations created by the proposed method.

Weaknesses:
1. The reviewer finds the manuscript hard to follow, especially Section II. The authors may come up with a clearer presentation.
2. The descriptions about saliency maps are less relevant to the main idea, further confounding the reviewer.
3. The comparison between simple gradient and direction derivative is less fair, as the directional derivative makes use of the very information direction w (e.g., the direction of no sunglass -> sunglass).  What happens if we visualize $\phi^{-1}(\phi(x)+ \alpha w)$ directly, for different values of $\alpha$.
4. The human study may need to conduct another set of control experiments to show that only original training images (not counterfactual interpolations) are $\textbf{less}$ helpful for uses to identify CNN patterns and biases.  The reviewer conjectures that for this simple TWO2TWO data, the subjects may spot shortcuts easily even using original training images.

Other minor comments:
1. Figure 1: There is no explanation for (a). What is $w$? The reader may not understand it for the first reading.
2. Figure 4: The reviewer believes normalized scores on the top of the images make better sense.

---

> ### Author Response · Authors · 2020-11-25
> **Answer R4**
>
> R4: Use of invertible CNNs to construct counterfactuals and isosurfaces
> > The reviewer finds the manuscript hard to follow [...]
>
> We agree with your criticism and reworked the entire manuscript substantially.
>
>
> > The descriptions about saliency maps are less relevant to the main idea [...]
>
> We agree that the description had ample room for improvement and we invested much time in focusing the text for improved clarity (we hope you agree).
>
> > The comparison between simple gradient and direction derivative is less fair, as the directional derivative makes use of the very information direction [...]
>
> We compare all methods on the same model running the same integration. The gradient and the directional derivative d\phi^{-1}/dw make both use of the direction w, as you can write both as the Jacobi Matrix J * w  and J^{-1} w. The reason for their different results is that J^{-1} is suited to translate w to image space and J is not. We therefore think the comparison is fair. However, we have rewritten the description of the method (sec. 3) and hope this improved clarity.
> If you visualize \phi^{-1}(\phi(x) + a w) directly, you will get the same result as when integrating d\phi^{-1}/dw over the respective length.
>
> > The human study may need to conduct another set of control experiments to show that only original training images (not counterfactual interpolations) are helpful for uses to identify CNN patterns and biases. [...]
>
> We have implemented this in the baseline condition: users are presented with images from the validation set, sorted by their logits. Since the study was a between-group design with each group only seeing one explanation technique, we can isolate the effect of studying only images and no interpolation. We concluded that studying examples this way is indeed helpful, but also does not allow more than half of the participants to detect the subtle rotation bias. This rejects the theory that users can find shortcuts easily on this dataset. While it is still a relatively simple dataset, it still allowed us to create user tasks that are not trivial. The writing in that section was very condensed and lacked clarity. We improved the text and have integrated your comment in the new version.
>
> > [Other minor comments] Figure 1: There is no explanation for (a). What is w The reader may not understand it for the first reading.
>
> Thanks for pointing this out, we have improved the caption.
>
> > [Other minor comments] Figure 4: The reviewer believes normalized scores on the top of the images make better sense.
>
> Do you mean not the raw logit value but rather the probability? We think the logit value provides a better summary as a change from 5 to 50 corresponds to a probability from 0.993 to 1-1e-22 which can be better understood as logit.

---

### Official Review · AnonReviewer1 · 2020-10-30
**Relevant. Lacks clarity. Mildly convincing results.**

**Rating:** 6
**Confidence:** 4

**Review:**

In this paper, the authors propose a method for generating counterfactuals (visually “similar” examples with different labels) and isofactuals (visually “different” examples with the same label) using an invertible convolutional network. A human study shows that providing these counterfactual and isofactual images in a systematic way can help participants understand a model’s bias better.

Interpretability of machine learning models is becoming progressively more important as these models continue to proliferate in sensitive applications such as medicine, finance, law, etc. A large body of interpretability efforts is around post hoc methods where an explanation is generated given certain probes into a blackbox function. On the other hand, the proposed model imposes certain constraints on the model to yield these explanations. The paper has some interesting ideas, and the qualitative evaluation is helpful is conveying those ideas. The discussion around gradient wrt the input image vs directional derivative is fairly insightful with convincing qualitative and quantitative results (Fig. 4). However, I did have some conerns:

1. The writing often lacks clarity and the usage of space can be more judicious. For example, the description of the main method is severely lacking, and is lumped into a few short paragraphs (section 2). I needed to reread the section a few times to understand the gist since important details are either missing or relegated to the appendix. I am still unsure about the development of isosurface section. On the other hand, excessive details are present in the evaluation section, e.g. in-depth discussion of mice characteristics. Focusing on elaborating the method, and maybe one less dataset would improve readability by moving extra evaluation to appendix. Something like, “we observe similar patterns with other tested datasets, which are presented in the appendix.”

2. The results of the human subject study are not very convincing. While the subjects were able to better detect model’s biases with the systematic presentation using the proposed method, they also spuriously discovered irrelevant patterns (background and blocks). The appendix (Fig. 9) also shows that the subjects thought both the proposed method and the baseline were equally good. [Digression: Fig. 9 is poorly processed with missing words, repeated legend, shuffled axes, etc.).

3. Minor formatting issues: mismatched quotes throughout (striped, zebra, etc.), Fig. 4 caption (there not ideal), “different to the original”, etc.

Overall, the paper has some good ideas and interesting analysis but falls short on clarity and fully convincing the reader about the results. I am marginally inclined for it to be accepted.

---

> ### Author Response · Authors · 2020-11-25
> **Answer R1**
>
> > The writing often lacks clarity and the usage of space can be more judicious. [...]
>
> We have substantially reworked the writing for better balance. We have expanded the methods and rewritten the isosurface section to improve clarity.
> > Focusing on elaborating the method, and maybe one less dataset would improve readability by moving extra evaluation to appendix. [...]
>
> We followed your advice and moved large parts of the mice evaluation to the appendix.
> > The results of the human subject study are not very convincing. [...]
>
> Unfortunately, the labels in the user study figure were swapped in the original version of the manuscript which may have led to the impression that participants find a lot of irrelevant patterns. We apologize and hope that the corrected figure and improved description clarifies our findings.
>
> > The appendix (Fig. 9) also shows that the subjects thought both the proposed method and the baseline were equally good.[...]
>
> We improved the layout of the figures and corrected the mistakes you mentioned. Regarding the results of the user study, both methods received comparable ratings. However, both ratings were considerably high and show that we were successful in designing both a usable counterfactual generation method and a baseline that users found useful as well. We discussed the implications of this result in the conclusions of the updated manuscript. On a sidenote: we preregistered our experiment and reported a standardised subjective ratings scale. It would be great if this (in ML rather non-standard) practice would be adopted in interpretability research because it shows that we care about rigorous evaluation with those that are supposed to use explanations: the users.
>
> > Minor formatting issues: mismatched quotes throughout (striped, zebra, etc.),[...]
>
> Thank you, we fixed all of them.
>
> > Overall, the paper has some good ideas and interesting analysis[...]I am marginally inclined for it to be accepted.
>
> Thank you. We agree with your criticism and reworked the entire manuscript substantially. We hope you agree this merits an updated rating :)

---

### Official Review · AnonReviewer5 · 2020-11-09
**While technically incremental, the work provides interesting method to generate multiple kinds of explanations counterfactuals, saliency maps and what is called as "isosurface" of the classifer. Interesting case-study and user study demonstrate potential benefit of the method comparing two types of explanations**

**Rating:** 6
**Confidence:** 4

**Review:**

1. The authors propose a method to derive counterfactuals, saliency maps, and so called isofactuals using invertible neural networks. The quality of the generated explanations are compared visually with existing baselines.

2. I found the structure of the paper confusing and lacking in clear elicitation of contributions. For example, once explanation methods are provided in Sec 2, the motivation of Section 3 to purely compare gradient methods is highly unstructured. In here, its unclear why changes in independent principal components while generating counterfactuals not desirable especially if as the authors suggest, the prediction changes. Even if the changes are not observable to humans. This is a highly unusual aspect of the counterfactuals where the counterfactuals shouldn't just explain large changes to logits but changes to predictions as well.

3. The objective of each evaluation data-set and study should also be clearly outlined before proceeding to the details. Currently the paper leaves the reader to figure out the main contributions at the cost of hampering the paper's technical significance.

4. What is the goal of the user study? Why is the baseline any other method of generating counterfactuals but merely conditioned examples?

I strongly suggest restructuring the paper to fix above concerns and provide a clear justification for their experimental setup.

5. I also strongly recommend the authors to move away from evaluating their models against celebA labels such as "Attractive" which are rife with ethical concerns. I understand that those are the only options available in celebA but I would recommend using more neutral class labels for experiments if face dataset is an important part of their evaluations.

--------------------------------------------------------------------------------------------------------------------------------------------------------------------------------------
The authors have done a reasonable job at addressing my concerns and I have increased my score from 5 to 6.

---

> ### Author Response · Authors · 2020-11-25
> **Answer R5**
>
>
>
>
> > 2. I found the structure of the paper confusing and lacking in clear elicitation of contributions.
>
> We have substantially restructured the entire manuscript. The contributions are clearly stated at the end of the introduction. We moved Sec 2 to the end of the evaluation and provide appropriate motivation for it.
> > In here [Sec 3], its unclear why changes in independent principal components while generating counterfactuals not desirable especially if as the authors suggest, the prediction changes. Even if the changes are not observable to humans. This is a highly unusual aspect of the counterfactuals where the counterfactuals shouldn't just explain large changes to logits but changes to predictions as well.
>
> We agree with the definition put forward in (Wachter et. al, 2018) which has  the“closest possible world” requirement which states that counterfactuals should not change unrelated properties. If they do, it would be hard to tell which change was responsible for the change in prediction. Small and invisible changes do not provide the user with information.
>
>
> > 3. The objective of each evaluation data-set and study should also be clearly outlined before proceeding to the details. Currently the paper leaves the reader to figure out the main contributions at the cost of hampering the paper's technical significance.
>
> We have restructured this part of the manuscript and clarified this in the updated version of the manuscript.
> > What is the goal of the user study? Why is the baseline any other method of generating counterfactuals but merely conditioned examples?
> We have now stated the goal in the first paragraph in the corresponding section. We also added more detail about our reasoning for not considering other counterfactual methods or saliency maps and discussed this choice.
>
> > I also strongly recommend the authors to move away from evaluating their models against celebA labels such as "Attractive" which are rife with ethical concerns. I understand that those are the only options available in celebA but I would recommend using more neutral class labels for experiments if face dataset is an important part of their evaluations.
>
> We share these ethical concerns. We chose celebA exactly because it was already criticised due to its many shortcomings. We expected to be able to confirm and investigate biases that we knew existed and to emphasize the advantages of our method. We now state ethical concerns in the manuscript explicitly.

---

### Author Response · Authors · 2020-11-25
**General Rebuttal Answer**

We want to thank the reviewer for their time, effort and detailed feedback. Their comments helped to significantly improve this paper. We address each reviewer’s comment in-line and provide an overview here.

We generally found the reviews to be a fair assessment of our paper. The reviewers pointed out strengths of our paper such as the novelty (R2), combining a generative and discriminative model (R3), the comparison of the directional derivative dx/dw to the gradient (R2), directional derivatives for constructing counterfactuals (R4), using PCA to create isosurfaces (R4) and conducting a rigorous user study (R2, R4).

The main point of criticism was the quality of the presentation. The manuscript lacked clarity and was oftentimes confusing. Some reviewers raised concerns about the design and the results of our user study.

Based on the reviewers' feedback, we rewrote large parts of the paper. We now state our motivations and contributions clearer. We restructured  the method section, adding clearer definitions for counterfactuals and ideal counterfactuals. We moved the comparison of the gradient with the directional derivative into the evaluation. For each dataset, we provide a justification of why we used it and how it fits into the overall evaluation of our method. Some parts were moved to the appendix to increase focus on the most important aspects and reduce clutter.  Additionally, improved the related work section mentioning how related methods differ from our approach. We think this improved clarity throughout the paper.

With respect to the evaluation, we added an analysis to the celebA section where we now report results of correlations that confirm different hypotheses. Regarding the user study, we found and fixed a severe mistake in a figure that was probably one cause of confusion, heartfelt apologies! Some reviewers criticised that our methods could not beat the baseline. While this is true, we want to emphasize that 1) conducting a user study itself is a valuable contribution that most papers in the field are still hesitant conducting, 2) both methods (our conterfactual and the baseline) have provided users with sufficient information to discover relevant and irrelevant features. This just indicates that our baseline was strong and we discuss this result and it’s potential implications for interpretability research in the conclusions of our manuscript.

We hope our substantial rework finds your liking, some additional results will find their way into the manuscript until the camera-ready version is due. If you agree that these changes improved our manuscript, we would be grateful for an increased rating.

---

### Decision · Program_Chairs · 2021-01-07
**Final Decision**

**Decision:**

Reject

**Comment:**

All the reviewers agree that the paper presents an interesting idea, and the main concern raised by the reviewers was the clarity of the paper. I believe that the authors have improved the presentation of the paper after rebuttal, however, I still believe that the paper woudl require another round of reviews before being ready for publication, in order to properly assess its contributions.